# Porous Polymer-Titanium Dioxide/Copper Composite with Improved Photocatalytic Activity toward Degradation of Organic Pollutants in Wastewater: Fabrication and Characterization as Well as Photocatalytic Activity Evaluation

**Qijie Xu [1,*], Yan Wang [2], Mei Chi [1], Wenbin Hu [1], Ning Zhang [1] and Weiwei He [3,*]**

[1]    College of Chemistry & Pharmaceutical Engineering, Huanghuai University,
       Zhumadian 463000, Henan, China; Chi820820@163.com (M.C.); hwb4449396@126.com (W.H.);
       ningzhang@126.com (N.Z.)
[2]    Institute of Science and Technology, Henan University, Kaifeng 475004, Henan, China;
       wangyan8079@henu.edu.cn
[3]    Key Laboratory of Micro-Nano Materials for Energy Storage and Conversion of Henan Province, Institute of
       Surface Micro and Nano Materials, College of Advanced Materials and Energy, Xuchang University,
       Xuchang 461000, Henan, China
*    Correspondence: qijie001@163.com (Q.X.); heweiweixcu@gmail.com (W.H.)

**Abstract:** Titanium dioxide ($TiO_2$) and $TiO_2$/copper (denoted as TC) composite were prepared via hydrothermal process. In the meantime, divinylbenzene (DVB) and bismaleimide (BMI) monomers were allowed to participate in in-situ radical polymerization in the presence of azobisisobutyronitrile (AIBN) initiator to afford porous polymers (abridged as PP). The as-obtained PP were mixed together with tetrabutyl titanate (TBT) and $CuSO_4 \cdot 5H_2O$ in vacuum to obtain PP/TC composite (denoted as PPTC) containing incorporated TC composite in the pores of PP. The as-prepared $TiO_2$, TC, and PPTC were characterized by transmission electron microscopy, scanning electron microscopy, X-ray diffraction, fluorescence spectrometry, and electron spin resonance spectrometry, and so on. Furthermore, their photocatalytic activity for the degradation of *N,N*-dimethylformamide, methyl orange, phenol, and methylene blue under the irradiation of simulated sunlight (Xe lamp light) and natural sunlight were investigated. Findings indicated that, whether under simulated sunlight or nature sunlight irradiation, PPTC exhibited much better photocatalytic performance than $TiO_2$ and TC for the degradation of the tested organic pollutants. Particularly, it allowed *N,N*-dimethylformamide (DMF) to be degraded by a rate of 73.7% under simulated sunlight irradiation and it retained photocatalytic activity even after six cycles of reuse, exhibiting promising potential for the removal of organic pollutants in wastewater (including industrial water, aquaculture wastewater, and domestic sewage). The desired photocatalytic performance of the as-prepared PPTC is attributed to two aspects. Namely, the incorporation of $Cu^{2+}$ into the fine structure of $TiO_2$ contributes to increasing photocatalyst activity and producing more free radical while the embedding of TC composite into the PP pores improves to the contact area between the photocatalyst and organic pollutants, and both are beneficial for improving the adsorption capacity and activity of the photocatalyst, thereby enhancing the degradation of the organic pollutants.

**Keywords:** porous polymer; $TiO_2$/Cu composite; photocatalyst; characterization; wastewater; organic pollutant; degradation; photocatalytic activity

## 1. Introduction

The photocatalytic degradation of organic pollutants on semiconductor materials is of particular significance for wastewater treatment, since the photocatalytic degradation is often realizable under mild conditions. In this sense, traditional semiconductor materials such as $TiO_2$, ZnO, and $Ag_3PO_4$ are often utilized as the photocatalysts, due to their low cost, good stability, non-toxicity, and desired environmental acceptance [1–4]. Among them, $TiO_2$ is a representative photocatalyst widely used for the photocatalytic degradation of organic compounds and water splitting to produce hydrogen [1].

The large-scale application of $TiO_2$ photocatalyst, however, is limited by three drawbacks. Firstly, $TiO_2$ is only responsive to ultraviolet light (3%~5% of sunlight) and has a low utilization efficiency of sunlight. Secondly, $TiO_2$ powder is difficult to separate from the reaction system for reusing the degradation system, and the photocatalyst in combination with inorganic materials used for separation has good access to water medium to cause secondary pollution. Fortunately, some $TiO_2$ based thin layers/films can achieve good separation from water [5,6]. Thirdly, $TiO_2$ photocatalyst has a high recombination rate of electron-hole pairs, which seriously affects its photocatalytic efficiency [7]. For getting rid of the above-mentioned drawbacks, some researchers conducted selective metal doping to improve the photocatalytic activity of $TiO_2$ and enhance its response in the visible-light region so as to acquire increased photocatalytic efficiency. Some of representative research in this aspect point to the preparation and photocatalytic activity evaluation of a series of titania-metal composites doped with transition metals [8,9], noble metals [10–12], and nonmetals [13–15].

Previous studies demonstrate that cupric ion is suitable for doping various visible light-responsive photocatalysts [16,17], since Cu element is of relatively low cost and abundant origin as well as low potential. More important, the two kinds of copper cation, $Cu^{2+}$ and $Cu^+$, can comprise $Cu^{2+}/Cu^+$ and $Cu^{2+}/Cu$ electrodes with a redox potential of 0.16 V and 0.52 V, respectively. This means that the electrons ($e^-$) generated upon the photo-excitation of photocatalyst can be directly trapped by $Cu^{2+}$, thereby leaving the holes with positive charge ($h^+$). The as-generated holes can escape from the oxidative valence of the photocatalyst to degrade organic compounds. In this way, metal ion-doped semiconductor materials can provide semiconductor-based photocatalysts with charge trapping sites, thereby reducing the electron–hole pair recombination rate of the latter and extending their light response range [17]. Even so, unfortunately, the metal ion-doped semiconductor-based photocatalysts still face challenge of separation from the photocatalytic reaction system and are accessible to the water system, thereby causing secondary pollution.

Bearing those considerations in mind, here we attempt to design a novel photocatalyst consisting of $TiO_2$/Cu composite (abridged as TC) and porous polymer. We focus on porous polymers (PP) linked by covalent-organic frameworks, because they exhibit excellent adsorption performance toward gases and liquids as well as good photo activity and chemical stability, while they can be easily separated from photocatalytic reaction system. This article reports the preparation of PP/$TiO_2$ composites doped by $Cu^{2+}$ (denoted as PPTC). It also deals with the photocatalytic performance of the as-prepared $TiO_2$, TC, and PPTC toward the degradation of *N,N*-dimethylformamide (DMF), methyl orange, phenol, and methylene blue under the irradiation of simulated sunlight (Xe lamp light) and natural sunlight.

## 2. Results and Discussion

### 2.1. Formation of PP and PPTC

Figure 1 schematically shows the processes for preparing PP and PPTC as well as their use for photocatalytic degradation of various organic pollutants. Briefly, DVB and BMI monomers are initiated by AIBN to participate in radical polymerization reaction, thereby affording PP. Then TBT and $CuSO_4$ are adsorbed into the pores of PP, followed by hydrothermal reaction forming $TiO_2$/Cu composite (i.e., TC) in the pores, thereby yielding PPTC with stable structure and excellent adsorbing performance. The as-obtained PPTC is further used as the photocatalyst to degrade organic pollutants such as DMF,

methyl orange, phenol, and methylene blue under the irradiation of simulated sunlight or nature sunlight, thereby even achieving mineralization.

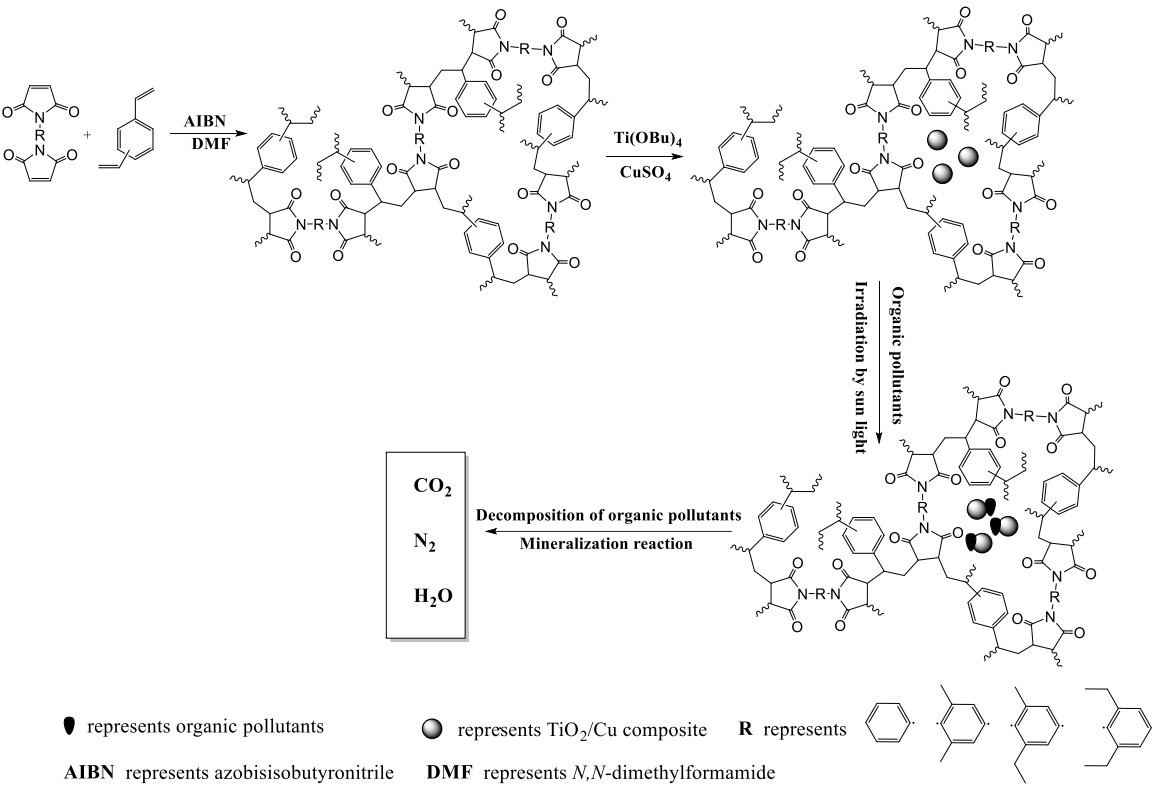

**Figure 1.** Schematic diagram for preparing ketch porous polymers and PPTC.

## 2.2. Characterization of TiO₂, TC, and PPTC

The SEM images of the as-prepared $TiO_2$, TC, and PPTC are shown in Figure 2. It can be seen that the as-prepared $TiO_2$ nanoparticles exhibit uneven sizes and tend to form aggregates (Figure 2A). The doping of $Cu^{2+}$ leads to some change in the fine structure of $TiO_2$ while the doped $Cu^{2+}$ tends to be gradually adhered onto the surface of $TiO_2$ matrix (Figure 2B). After $TiO_2$/Cu nanocomposite is incorporated into the pores of PP via in-situ hydrothermal process, the resultant PPTC exhibits uniform nanoporous structure and the incorporated $TiO_2$/Cu composite is free of aggregation (Figure 2C). Such a nanoporous structure of PPTC is favorable for increasing the contact area between the photocatalyst and organic pollutants, thereby improving the adsorption capacity and photocatalytic activity of the photocatalyst [8,11,14,15].

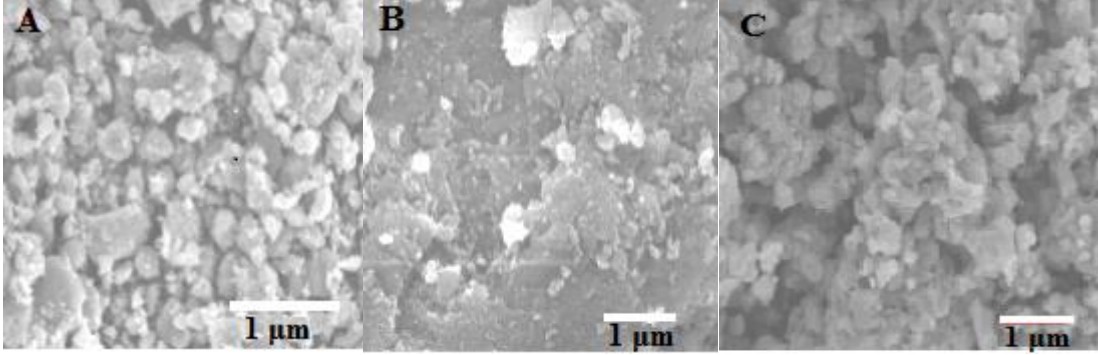

**Figure 2.** SEM images of (**A**) TiO₂, (**B**) TC, and (**C**) PPTC.

Figure 3 displays typical TEM micrographs of TiO$_2$, TC, and PPTC. It is seen that TiO$_2$ has an irregular structure (Figure 3A). Cu nanoparticles with size of approximately 10 nm are doped into the structure of TiO$_2$ and prevented from aggregation in the as-formed TC composite (Figure 3B), and TC is embedded into the pores of PP to form PPTC while the structure of TiO$_2$ nanoparticle is kept nearly unchanged (Figure 3C). Moreover, the TC nanoparticles exhibit good dispersion in the as-prepared PPTC composite, which is because PP with a nanoporous structure can hinder their aggregation therein well.

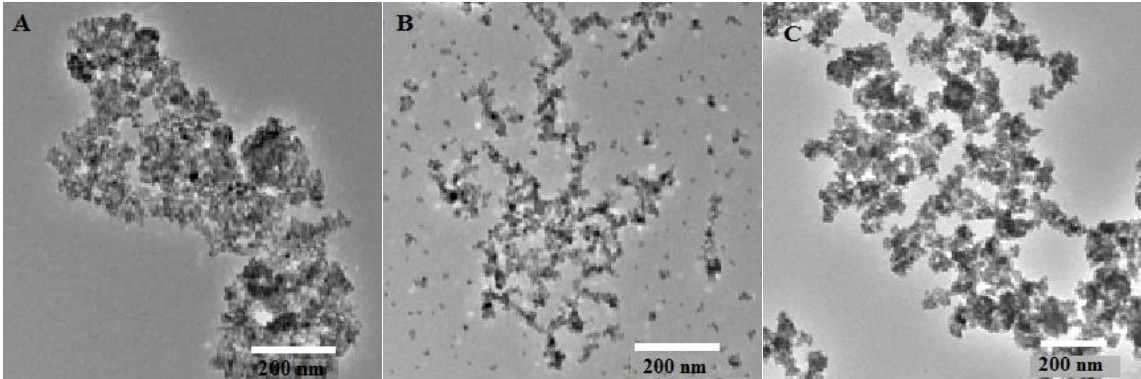

**Figure 3.** TEM images of (**A**) TiO$_2$, (**B**) TC, and (**C**) PPTC.

Figure 4 shows the XRD patterns of TiO$_2$, TC, and PPTC. The XRD peaks around $2\theta$ = 25.3°, 38°, 47.7°, 54.8°, 62.8°, and 68.8° correspond to the (101), (004), (200), (105), (104), and (200) crystal faces of nano-TiO$_2$, respectively [5–7], and they prove that the as-prepared TiO$_2$ nanoparticle is of an anatase structure. The two peaks at about 43.5° and 50.6° possibly correspond to metallic Cu of TC [8,18]. IN addition, doping Cu leads to little change in the crystal structure of TiO$_2$, and the TiO$_2$/Cu composite embedded in the pores of PP via calcination of TBT and CuSO$_4$·5H$_2$O in vacuum retains the initial state [16,17], which could be beneficial for enhancing the contact area between TiO$_2$/Cu composite and organic compound. Furthermore, we calculate the crystallization size of PPTC, TC, and TiO$_2$ from the XRD by using Scherrer formula and the sizes are 7.19 nm, 8.01 nm, and 7.22 nm, respectively. Thus, these results are almost equal to that of TEM. Moreover, the as-prepared PPTC do not obviously show the XRD signal of Cu, possibly because their content of doped Cu is too low, furthermore, that was covered by PP.

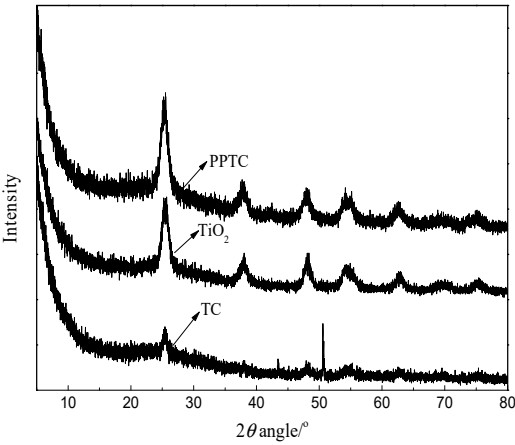

**Figure 4.** XRD patterns of TiO$_2$, TC, and PPTC.

Figure 5 illustrates the UV-vis absorbance spectra of TiO$_2$, TC, and PPTC. There are great differences in the optical absorbance behaviors of TiO$_2$, TC, and PPTC. On the one hand, the gradient decrease of

the reflectance at approximately 277 nm after the incorporation of Cu could be attributed to surface plasmon resonance that leads to enhanced photon capture at the visible region that is obviously stronger than $TiO_2$ [19]. On the other hand, TC has much stronger visible light absorbance than $TiO_2$ while PPTC shows broader photoresponse range as well as stronger visible light absorbance at about 425 nm as compared with $TiO_2$ and TC. This indicates that the embedding of TC in the pores of the PP gives rise to PPTC photocatalyst with broadened light response range and enhanced photocatalytic performance toward the degradation of organic pollutants [19,20].

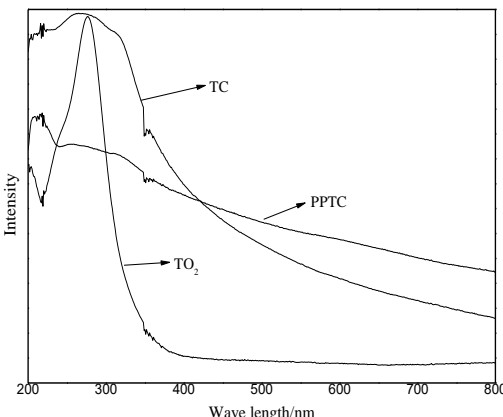

**Figure 5.** UV-vis absorbance spectra of $TiO_2$, TC, and PPTC.

## 2.3. Photocatalytic Performance of $TiO_2$, TC and PPTC

The photocatalytic degradation rates of DMF catalyzed by $TiO_2$, TC, and PPTC under simulated sunlight irradiation for different duration (0.5~3.0 h) are shown in Figure 6. It can be seen that $TiO_2$, TC, and PPTC as the photocatalysts all can catalyze the photocatalytic degradation of DMF under simulated sunlight irradiation, and the degradation rate of DMF gradually increases with increasing irradiation time thereunder. Particularly, after 3 h of illumination in the presence of PPTC, TC, and $TiO_2$, the tested DMF is degraded by a rate of 73.7%, 65.4%, and 52.8%, respectively. This demonstrates that the as-prepared PPTC photocatalyst possesses excellent photocatalytic degradation efficiency toward DMF.

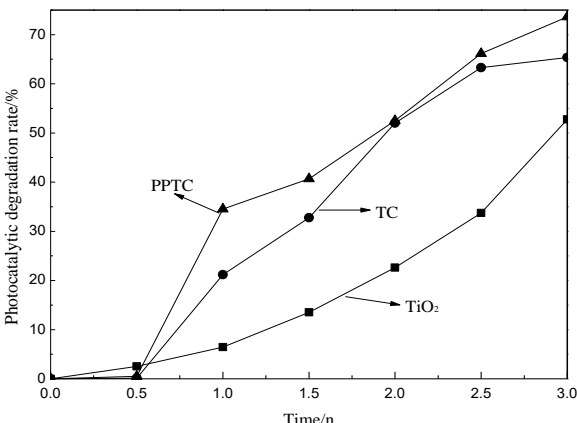

**Figure 6.** Photocatalytic performance of $TiO_2$, TC, and PPTC toward degradation of *N,N*-dimethylformamide (DMF).

Usually, the photocatalytic performance of photocatalyst is highly dependent on its specific surface area and the recombination rate of its holes and photo-excited electrons [20]. It has been found that the enhancement in the adsorption capacity of photocatalysts contributes to increasing photocatalytic efficiency [7,11]. As listed in Table 1, PPTC has the largest specific surface area (whether by the

BET (Brunauer, Emmett, and Teller) multi-point-contact method or by Langmuir single-point-contact method) as well as the maximum total pore volume of single-point adsorption among the three tested photocatalysts, which is why it exhibits better photocatalytic performance than $TiO_2$ and TC toward the degradation of DMF [20,21]. In other words, as compared with $TiO_2$ and TC, PPTC with larger specific surface area and total pore volume as well as smaller average pore size would have more opportunities to contact and adsorb DMF, thereby achieving enhanced photocatalytic degradation of the organic pollutant [21,22].

**Table 1.** Specific surface area of $TiO_2$, TC, and PPTC.

| Photocatalyst | Brunner Emmet Teller (BET) Multi-Point Contact ($m^2/g$) | Langmuir Single-Point Contact ($m^2/g$) | Total Pore Volume of Single-Point Adsorption ($cm^3/g$) | Average Adsorption Pore Size (nm) |
|---|---|---|---|---|
| $TiO_2$ | 61.9 | 93.7 | 0.3 | 19.5 |
| TC | 385.4 | 594.7 | 0.4 | 4.7 |
| PPTC | 486.7 | 693.6 | 0.5 | 4.1 |

In the meantime, as shown in Figure 7, $TiO_2$ has the strongest emission intensity while PPTC has the lowest one. This reveals that PPTC has a low hole-electron recombination rate, which corresponds well to its enhanced photocatalytic degradation ability for DMF [21].

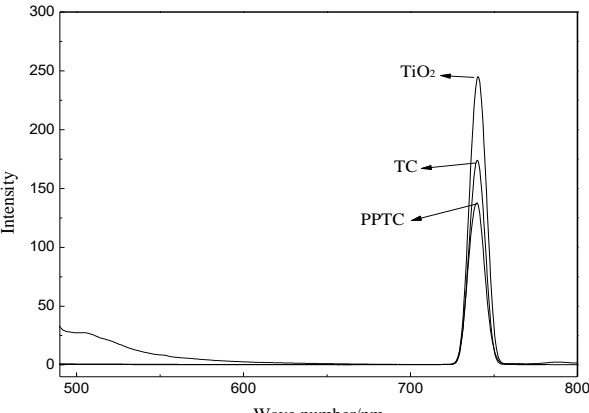

**Figure 7.** Fluorescence spectra of $TiO_2$, TC, and PPTC.

The recyclability of $TO_2$, TC, and PPTC toward the photocatalytic degradation of DMF is shown in Figure 8. $TiO_2$ and TC can be reused four times and five times, respectively. Photocatalyst PPTC, however, is still able to degrade DMF even after six cycles of reuse. There is obvious difference in the three photocatalysts that PPTC can be reused more times in photocatalytic degradation of DMF. This could be because the $TiO_2/Cu$ composite is embedded well in the pores of the porous polymer and prevented from loss upon exposing to simulated or natural sunlight irradiation, thereby leading to improved recyclability of PPTC than $TiO_2$ and TC [22]. Regrettably, PPTC only has about 20% degradation rate after five times of reuse, the main reason may be attributed to organic pollutants covering the pore of photocatalysts [23]. In addition, the little leaching of metallic Cu also affects the activity of photocatalysts.

Table 2 lists the photocatalytic degradation rate of DMF catalyzed by $TiO_2$, TC, and PPTC without illumination (in the dark) and under the irradiation of simulated (Xe lamp) or real sunlight. Without illumination, the degradation rate of DMF is very low even under the catalysis by PPTC, possibly due to highly retarded adsorption of DMF thereunder. Besides, whether under simulated sunlight or nature sunlight irradiation, the degradation rate of DMF catalyzed by PPTC is the maximum (73.7% and 64.0%), which indicates that PPTC could have good prospects for the photocatalytic degradation of organic pollutants like DMF, methyl orange, methylene blue, phenol, and other organic pollutants.

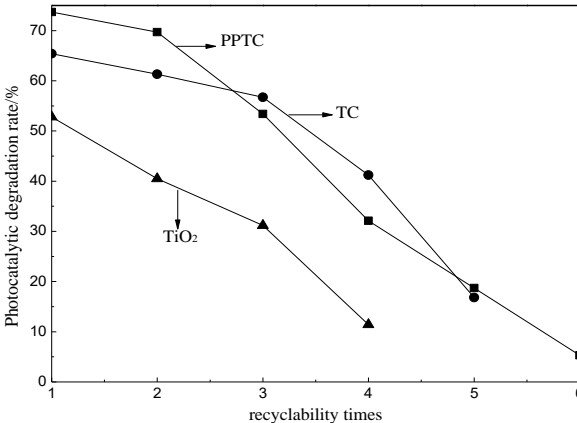

**Figure 8.** Photocatalytic degradation rate of DMF as a function of recycling number of $TiO_2$, TC, and PPTC.

**Table 2.** Degradation rate of DMF catalyzed by photocatalysts without illumination and under irradiation of simulated sunlight (Xe lamp) or real sunlight.

| Photocatalyst | Degradation Rate of DMF | | |
| --- | --- | --- | --- |
| | No Light | Xenon Light | Sunlight |
| PPTC | 26.0% | 73.7% | 64.0% |
| TC | 21.2% | 65.4% | 57.6% |
| $TiO_2$ | 6.5% | 52.8% | 48.3% |

Usually, the photocatalyst with a high adsorption capacity often has more opportunities to contact the organic pollutant, and the degradation of the organic pollutant is closely related to the adsorption capacity of the photocatalyst. Table 3 summarizes the photocatalytic degradation rates of several organic compounds under the catalysis of $TiO_2$, TC, and PPTC without illumination and under the irradiation of simulated sunlight (Xe lamp). It can be seen that the three photocatalysts can all photocatalyze the degradation of DMF, methyl orange, methylene blue, and phenol well, and PPTC has better photocatalytic activity than $TiO_2$ and TC. Particularly, PPTC can adsorb the organic pollutants more efficiently than $TiO_2$ and TC, which corresponds well to its better photocatalytic activity for the degradation rate of the tested organic pollutants. This could be because PPTC exhibits larger specific surface area and total pore volume than $TiO_2$ and TC, which corresponds to an increased contact area with the organic pollutants [22,24]. Further research in this respect is underway.

The photocatalytic performance of a photocatalyst is highly dependent on its light response range and adsorption capacity that are related to free radical intensity [25–27]. Here we measure the ESR spectra of various photocatalysts to estimate the intensities of superoxide free radical ($O_2^{\cdot}$), singlet oxygen ($^1O_2$), and hydroxyl radical ($\cdot OH$), hoping to better understand the photocatalytic mechanisms of the as-prepared photocatalysts. As shown in Figure 9, the superoxide free radical intensities of TC and PPTC are nearly identical and higher than that of $TiO_2$, which indicates that doping $Cu^{2+}$ into $TiO_2$ increases the activity of the photocatalyst and improves its photocatalytic ability for the degradation of the organic pollutants. The $H_2$-TPR results of Figure 10 reveal the reduced temperature of the Ti element from $TiO_2$ by hydrogen is clearly higher than those of TC and PPTC. This could be explained by the supposition that $Cu^{2+}$ ions doped into $TiO_2$ increase the activity of the photocatalyst and produce free radicals, as evidence by relevant ESR results. Moreover, the incorporation of the TC composite into the pores of PP contributes to an increasing contact area between the photocatalyst and organic pollutants, thereby improving the adsorption capacity of the photocatalyst and enhancing the degradation of the organic pollutants (see Table 3).

**Table 3.** Photocatalytic degradation rates of several organic pollutants catalyzed by TiO$_2$, TC, and PPTC in the dark and under the irradiation of Xe lamp light.

| No. | Photocatalyst | Organic Pollutant | Adsorption Rate/% [a] | Degradation Rate/% [b] |
|---|---|---|---|---|
| 1 | TiO$_2$ | DMF | 6.5 | 52.8 |
| 2 | TC | DMF | 21.2 | 65.4 |
| 3 | PPTC | DMF | 26.0 | 73.7 |
| 4 | TiO$_2$ | Methyl orange | 3.6 | 55.5 |
| 5 | TC | Methyl orange | 29.7 | 91.6 |
| 6 | PPTC | Methyl orange | 53.9 | 97.8 |
| 7 | TiO$_2$ | Methylene blue | 40.0 | 90.9 |
| 8 | TC | Methylene blue | 66.3 | 94.3 |
| 9 | PPTC | Methylene blue | 94.9 | 100 |
| 10 | TiO$_2$ | Phenol | 32.1 | 83.6 |
| 11 | TC | Phenol | 46.3 | 90.7 |
| 12 | PPTC | Phenol | 58.7 | 98.6 |

Note: "a" represents the adsorption of the organic compounds by photocatalysts in the dark environment; "b" represents the photocatalytic degradation rate of the organic compounds catalyzed by the photocatalysts under the irradiation of Xe lamp light.

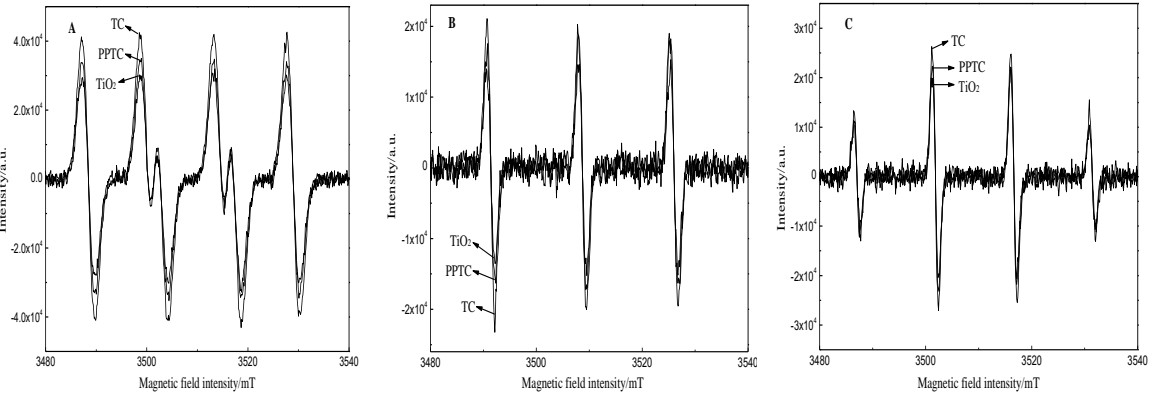

**Figure 9.** ESR spectra of (**A**) superoxide free radical (O$_2^{\cdot}$), (**B**) singlet oxygen ($^1$O$_2$), and (**C**) hydroxyl radical (·OH) in TiO$_2$, TC, and PPTC.

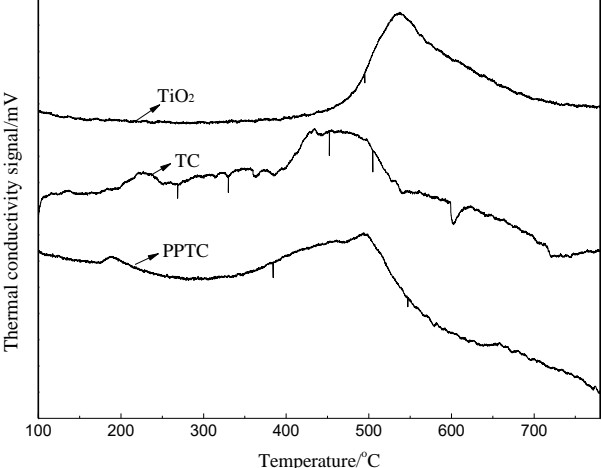

**Figure 10.** H$_2$-temperature-programmed reduction (TPR) curves of TiO$_2$, TC, and PPTC.

## 3. Experimental

### 3.1. Chemicals

Divinylbenzene (DVB) and azobisisobutyronitrile (AIBN) were supplied by Shanghai Macklin Biochemical Company Limited (Shanghai, China). Bismaleimide (BMI) was obtained from Huading Company Limited (Zhumadian, China). Tetrabutyl titanate (TBT) was supplied by Tiancheng Chemicals (Shandong, China). $CuSO_4 \cdot 5H_2O$ was purchased from Tianjin Kaitong Chemicals (Tianjin, China), and *N,N*-dimethylformamide (DMF) was supplied by Sinopharm Chemicals Reagent Company Limited (Beijing, China). All the reagents used in this work are of analytical grade.

### 3.2. Preparation of TiO₂, TC, and PPTC

$TiO_2$ and TC were readily prepared via hydrothermal synthesis [11]. Briefly, 2 mL of TBT was added into 20 mL of absolute ethanol in a three-necked flask and magnetically stirred with a magnet for 30 min. The resultant solution was transferred to a 30 mL hydrothermal reactor and heated at 210 °C for 24 h. Upon completion of reaction, the product was washed thrice with distilled water and absolute ethanol and heated at 80 °C for 24 h to afford $TiO_2$ powder. TC was obtained in the same manner while a proper amount of $CuSO_4 \cdot 5H_2O$ solution was simultaneously introduced into the hydrothermal reactor together with the ethanol solution of TBT. The as-prepared TC contains 1.0% (atomic ratio) of doped $Cu^{2+}$ ions.

The in-situ radical polymerization of DVB and BMI monomers in the presence of AIBN initiator was conducted to afford PP, and the as-obtained PP were mixed together with TBT and $CuSO_4 \cdot 5H_2O$ in the same procedures of TC, where as-formed TC was embedded into the pores of PP thereby affording desired PPTC. In brief, 0.02 mol of DVB, 0.02 mol of BMI, 0.2 mmol of AIBN, and 20 mL of DMF were sequentially added to a 250 mL three-necked flask and stirred at 80 °C for 6 h under Ar atmosphere to afford PP after the product was washed with absolute ethanol. Into the ethanol solution of PP were then added 2 mL of TBT and a proper amount of $CuSO_4 \cdot 5H_2O$ solution. The resultant mixed solution was transferred to the hydrothermal reactor and heated at 210 °C for 24 h. At the end of reaction, the product was filtered, washed with methanol, and dried at 80 °C in a vacuum oven to obtain PPTC.

### 3.3. Characterization and Photocatalytic Activity Evaluation

A transmission electron microscope (TEM, JEM-2010, JEOL Company, Tokyo Metropolitan, Japan) was performed at an accelerating voltage of 200 kV to observe the dispersion behavior of TC in the pores of PP and the pore size of PP. A scanning electronic microscope (SEM, JSM 5600LV, JEOL Company, Tokyo Metropolitan, Japan) was employed at an accelerating voltage of 30 kV to examine the microstructure of $TiO_2$, TC, and PPTC. Prior to SEM observation, the surfaces of the specimens were spray-coated with gold. The X-ray diffraction (XRD) patterns of the as-prepared $TiO_2$, TC, and PPTC were obtained with a D/max 2550 V X-ray diffractometer (Philips, Amsterdam, Holland; Cu $K_\alpha$ radiation, $\lambda = 1.54178$ Å). The ultraviolet-visible light (UV-vis) absorbance spectra of $TiO_2$, TC, and PPTC were measured over the wavelength range of 200~800 nm with a Cary 100 spectrophotometer. A fluorescence spectrometer (F-2700, Hitachi Company, Tokyo, Japan) was utilized to evaluate the fluorescent performances of $TiO_2$, TC, and PPTC in the wavelength range of 220~730 nm. $N_2$ physisorption of the as-fabricated samples were performed with a chemisorption surface area analyzer (JW-BK 222) at −196 °C for determining the BET (Brunauer, Emmett and Teller) surface area. Before the sorption measurements, the samples were outgassed at 200 °C for 1 h. The reduction behavior of $TiO_2$ and $Cu^{2+}$ in the photocatalysts and the interactions between them were examined by monitoring the temperature-programmed reduction of $H_2$ ($H_2$-TPR) with a hydrogen temperature programmed reduction instrument. X-band electron spin resonance (ESR) spectra were recorded at room temperature with an ERS-221 spectrometer to detect superoxide free radical ($O_2^{\cdot}$), singlet oxygen ($^1O_2$), and hydroxyl radical (·OH) under dark and irradiation, respectively. The operation of ESR as following: 20 mg of samples was dispersed in methanol of 10 mL by ultrasonic method, the above 200 μL solution was taken to mix with DMPO

(5,5-dimethyl-1-pyrroline N-oxide) solution (50 mM) of 200 μL, then the mixture solution was placed in a quartz sample tube (5 mm O. D.) with a Young vacuum joint and a stopcock, as evacuating by a rotary pump and closed.

The photocatalytic performances of $TiO_2$, TC, and PPTC were evaluated by monitoring the decomposition of DMF and the other organic pollutants under the irradiation of simulated sunlight and nature sunlight where the operation condition as following: simulated sunlight was got by using Xe lamp of 500 W (CHF-XM500, Perfect light) with electric current of 20 A and the distance of 30 cm between the breaker and the lamp holder, and those under nature sunlight was that the mixture solution of pollutants and photocatalyst was set in the outside platform. With DMF as a representative, 0.05 g of the photocatalyst and 50 mL of DMF solution (100 mg/L; solvent: water) were added into a 100-mL breaker. The resultant solution was magnetically stirred for 1 h to achieve adsorption saturation before illumination, followed by exposing to simulated or nature sunlight at room temperature (controlled with a constant-temperature circulating pump) for 0.5 h. Upon completion of irradiation, the aqueous solution (about 2 mL) in the reactor was transferred to a 10 mL test tube and centrifuged for about 30 min. The UV-vis absorbance spectrum of the as-centrifuged solution was recorded at a wavelength of 464 nm so as to estimate its content of residual DMF. The other organic pollutants (methyl orange, methylene blue, and phenol) were photocatalytically degraded in the same manner, and their UV-vis absorbance spectra upon completion of the photocatalytic degradation were also measured under the same condition to determine their residual content in the solution.

## 4. Conclusions

$TiO_2$ and $TiO_2$/Cu composite are readily prepared via hydrothermal reaction with TBT and TBT + $CuSO_4·5H_2O$ as raw materials. In the meantime, the PP obtained by the in-situ radical polymerization of DVB and BIM in the presence of AIBN initiator were mixed together with TBT and $CuSO_4·5H_2O$ in vacuum to obtain PPTC upon the incorporation of TC into the pores of PP. Characterizations reveal that the as-prepared PPTC exhibits uniform nanoporous structure and the TC composite incorporated in its pores exhibits good dispersion and is free of aggregation, which is favorable for improving its adsorption capacity and photocatalytic activity. In addition, the as-prepared PPTC has the best photocatalytic activity toward the degradation of DMF, methyl orange, methylene blue, and phenol under the irradiation of simulated sunlight and real sunlight. This is because doping $Cu^{2+}$ into $TiO_2$ increases the activity of the photocatalyst and produces superoxide free radicals while the incorporation of TC composite into the pores of PP contributes to increasing contact area between the photocatalyst and organic pollutants, both contribute to improving the adsorption capacity and activity of the photocatalyst and enhancing the degradation of the organic pollutants. Particularly, PPTC has the maximum photocatalytic degradation efficiency for DMF under Xe lamp light irradiation (73.7%) as well as reuse properties, showing promising potential for the elimination of organic pollutants in wastewater.

**Author Contributions:** Q.X. contributed to investigation, writing-review & editing; Y.W. contributed to investigation and editing; M.C. contributed to investigation; W.H. (Wenbin Hu) contributed to investigation; N.Z. contributed to development or design of methodology; creation of models; W.H. (Weiwei He) contributed to project administration and acquisition of the financial support for the project leading to this publication. All authors have read and agreed to the published version of the manuscript.

**Funding:** This research funded by the Zhongyuan Thousand Talents Project (204200510016), the Program for Innovative Research Team (in Science and Technology) in University of Henan Province (19IRTSTHN026), the project of youth backbone teachers of Henan province (Grant No. 2017GGJS172), the Science and Technology Research Project of Henan province (Grant No. 182102410090 and 192102310492), and the APC was funded by W.H. (Weiwei He).

**Acknowledgments:** Q.X. would like to thank the support of the key science and technology innovation demonstration projects of Henan province and 2018's open topic on key laboratory of nanoscale energy storage and conversion materials of Henan province in China (2018ISMNM01).

**Conflicts of Interest:** The authors declare no conflicts of interest.

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
