# Peer review of "Porous Polymer-Titanium Dioxide/Copper Composite with Improved Photocatalytic Activity toward Degradation of Organic Pollutants in Wastewater: Fabrication and Characterization as Well as Photocatalytic Activity Evaluation"

_catalysts, doi:10.3390/catal10030310_

Round 1

Reviewer 1 Report

This manuscript reports on photocatalytic properties of a porous polymer-titanium dioxide/copper composite. In this point, the contents of the present work is interesting. The study itself sounds good. However, few things needs to be addressed before my approval for publication. In detail:

General

There are several misspelled words and other typos throughout the whole manuscript. I suggest to the authors to thoroughly revise the manuscript and take care of such typos/mistakes.

 Introduction

The authors present a paper with the focus on photocatalysis. However, they do not cite “Fujishima and Honda, Naturevolume 238, pages37–38 (1972)” as a pioneering work. The reviewer believe that it is mandatory to cite this paper as it basically started the field of photocatalysis.

Page 2 line 50-52. Authors state that the separation of TiO2 from water is difficult. However, this is not completely true. Nowadays, there are numerous TiO2 based thin layers/films (e.g. nanotubes prepared via electrochemical anodization) which do not form hydrocolloids in water and are used for photoinduced air, water treatment and also with antimicrobial properties. I recommend to authors to state this and add appropriate citations e.g. Krbal et al. Applied Materials Today 17 (2019) 104-111; Sopha et al. Electrochemistry Communicatons 111 (2020); Motola et al. Chemical Papers 73 (2019) 1163-1172; Motola et al. 287 (2017) 59-64.

The reviewer believe that in a well-written scientific publication, such information needs to be included for the wide audience.

Results and Discussion

General – the authors should use the same colour of lines in all Figures. For example. In Fig. 8 TiO2 is black but in Fig. 9 is blue. Moreover, in Fig. 7, the authors don’t use any colours. I strongly recommend to use the colours and to use the same colours in every Figure throughout the manuscript.

How can one distinguish Cu particles in Fig. 2B? It would be beneficial to provide some EDX mapping to see the distribution of Ti, Cu, O, etc.

The authors should describe why the original TiO2 (which forms agglomerates) does not form agglomerates after Cu doping.

Fig. 4 XRD; What is the diffraction at around 50-51° 2 theta in TC? Moreover, did you calculate the particle size from the XRD? Is it comparable with the results obtained from TEM?

The authors state a good recyclability of their samples. However, based on Fig. 9 the photocatalytic activity strongly decreases. For PPTC the degradation rate is >70% but drops to >50% after 3rd usage and further decrease to ~20% after 5 uses. How do you explain such decrease in photocatalytic activity for all samples? In the case of Cu doped samples, do you expect metal leaching? The authors state that their samples contain 1%Cu, prior to photocatalytic measurements. Do you have the same concentration of Cu after photocatalytic measurements? Moreover, how did the authors estimate the concentration of Cu in TC and PPTC?

In my view, the study is interesting and worth of investigation. I would reconsider this manuscript for publication after the suggested revisions.

Author Response

Dear Reviewer

     We thanks very much for your suggestions and comments. Our paper “Porous polymer-titanium dioxide/copper composite with improved photocatalytic activity toward degradation of organic pollutants in wastewater: Fabrication and characterization as well as photocatalytic activity evaluation” (Ms. catalysts-723305) has been revised in response to the comments and suggestions.

     Unfortunately, A terrible novel coronavirus pneumonia is rapidly spread  in China in this month, all scientific research institutions and universities are closed, thus some datas, e.g. EDX, TEM can not be repeated in this period. I will provide these datas while I can go to work.

     The other revised introduction are in the enclosure file. 

Reviewer 2 Report

The manuscript presents the use of TiO2, TiO2/Cu and a porous polymer-titanium dioxide/copper composite in the degradation of several organic pollutants. Although the study is original the manuscript lacks of explanations and proper description of both experimental and characterization work. I do not recommend publishing the manuscript in this current form. 

The following points should be addressed before acceptance:

line 90: there is any reason for using a three-necked flask just for a simple magnetically stirring procedure?

line 102: should we understand that PP forms a solution with ethanol?

lines 94-103: as long as no patent application was specified in the manuscript, I see no reason for the authors not to specify exactly the preparation procedure for PP and PPTC. Thus, therms like "proper amounts" and "pre-set duration" should be replaced with numbers.

line 124: what Reakhim means and what about the adsorption of chlorobenzene? There is no data regarding this procedure within the manuscript.

lines 124-125: what photocatalyst have been "activated" at 500 C and why did the authors choose this temperature?

line 131: what "simulated sunlight" means? Did authors used any AM 1.5G filter for the Xe lamp? If it is not specified then the readers could assume that this is not a true simulated sunlight irradiation...it is just the light coming from a simple Xe lamp (with no specified intensity). Moreover, what about the solar light? Did the authors performed the reactions outside or inside the lab? If the sunlight went through a window then this is another story...(was the glass of the window covered with any protecting coating? what was the transmission? etc.). Thus, the authors should present exactly how they performed the photocatalytic experiments.

Fig. 1 contains typographical errors.

Fig. 3 The images are inconclusive. It is impossible for me to see any Cu nanoparticles or any diamond structure... Please add proper TEM images!

Fig. 4 Although the crystallinity of the samples is quit poor and the patterns are noisy the authors have identified "little change" in the case of TC... What about the peak at around 51 (2 theta value)?

Fig. 5 The scale bar is missing. Can the authors provide any references for the statements from lines 186-188 ("On the one hand....at the UV region")?

Fig. 6 There is any reason for performing the TG measurements?

Fig. 7 is wrong. Does make no sense to start from 100% degradation rate and to end after 3 h of irradiation at around 35%. The authors should check and correct the graph! 

Table 1 The authors didn't mention anything in the section 2.3 about the adsorption measurements.

lines 235-236 it is a bit too much to say that you have "excellent recyclability" considering that already after the third cycle the degradation rate drops to almost 30%.

Fig. 10 I see no big differences between the ESR spectra. Even a small difference in weight of the three materials can make this difference appear. Meanwhile, the authors do not mention how they performed these measurements. For instance, did the authors used any spin traps?

Author Response

Dear Reviewer

    We thanks very much for your suggestions and comments. Our paper “Porous polymer-titanium dioxide/copper composite with improved photocatalytic activity toward degradation of organic pollutants in wastewater: Fabrication and characterization as well as photocatalytic activity evaluation” (Ms. catalysts-723305) has been revised in response to the comments and suggestions.

    Unfortunately, A terrible novel coronavirus pneumonia is rapidly spread  in China in this month, all scientific research institutions and universities are closed, thus some datas, e.g. EDX, TEM can not be repeated in this period. I will provide these datas while I can go to work.

    The other revised introduction are in enclosure file.

Round 2

Reviewer 1 Report

Dear authors, it is unfortunate that your institutions and universities are closed due to coronavirus. I honestly hope that all the authors are safe and I sincerely wish a happy ending regarding the current situation in China (and worldwide). Nevertheless, few points needs to be addressed before acceptance.

  • The reviewer suggested to use the same colour of all lines in all Figures in the whole manuscript for better eye guidance and clearliness. (see Point 4 in the original comments to the authors). However, the authors did not provide such editing of the figures. The reviewer, once again, asks for such revision of all Figures.
  • The authors should provide EDX of all samples (point 5 in the original comments to the authors) after your institutions are open again.
  • Point 7 in the original comments to the authors. The authors needs to clearly define the diffraction at 50 -51° 2 Theta in Fig. 4. Moreover, they did not answer the other questions from Point 7.
  • The authors should address Point 8 in the original comments to the authors

Author Response

Dear Reviewer

    We thanks very much for your suggestions and comments. Our paper “Porous polymer-titanium dioxide/copper composite with improved photocatalytic activity toward degradation of organic pollutants in wastewater: Fabrication and characterization as well as photocatalytic activity evaluation” (Ms. catalysts-723305) has been revised again in response to the comments and suggestions.

  However, some testing data are still not done now, because a terrible coronavirus pneumonia has affected all scientific research institutions and universities in this period. I will provide these datas while I can go to work.

    The other revised introduction are in enclosure. 

Reviewer 2 Report

The revised form of the manuscript does not yet meet the requirements of a scientific article. Thus, once again, I do not recommend its publishing in this current form.

Given the current situation in China, I hope the publishers will give authors a chance to perform the required TEM experiments. However, before that, the authors should give more time to the following unsolved issues: 

line 92: The authors stated that "TiO2 and TC were readily prepared via hydrothermal synthesis[11]". However, in the reference 11 the materials were prepared by sol-gel method. Please add a proper reference!

lines 128-130: The ESR experimental procedure is still incomplete. For instance, the authors did not mention what kind of solvent was used and if they used any irradiation source and spin traps. Without these information it is impossible for me to understand how the authors make possible the detection of all these radicals (superoxide free radical, singlet oxygen and hydroxyl radical).

lines 131-143: The authors fail to explain how they exactly performed the reactions. Once again, the authors didn't provide any evidence that they used a "simulated sunlight" source (a simple picture of the equipment it is not enough). The authors might say at least 2 information: the name of the "simulated sunlight lamp" (the provider) and the intensity of this Xe lamp (measured at the distance where the beaker containing both the photocatalysts and the pollutants was placed). Moreover, it is still unclear in the case of natural sunlight irradiation if the reaction was performed outside laboratory or not.

Fig. 1: represents instead of "repesents"

lines 170-176: The authors must provide other TEM images in order to prove what they stated in the manuscript. Moreover, can the authors provide any references for the "diamond structure of TiO2 nanoparticles" formation? Actually, what this "diamond structure" means??

line 183: I do believe that there is only one XRD peak. I would change the sentence "Cu element" with a more appropriate one, like metallic Cu... Please check the literature for proper references.

lines 192-194: The statement "On the one hand, the gradient decrease of the reflectance at approximately 277 nm after the incorporation of Cu could be attributed to surface plasmon resonance that leads to enhanced photon capture at the UV region [18]" is not sustained by the reference [18]. Actually, the Cu surface plasmon resonance absorption can be detected in the visible region of the spectrum (at around 610 nm). The authors should clarify this statement or to provide a proper reference (if there is any in the literature...).

line 299: the sentence "excellent recyclability" should be changed.

General: the therm "composite" is more appropriate for the PPTC. TC is not a composite, it is just a doped material. Moreover, the sentence "stable complex" (line 235) should be erased.

Author Response

Dear Reviewer

    We thanks very much for your suggestions and comments. Our paper “Porous polymer-titanium dioxide/copper composite with improved photocatalytic activity toward degradation of organic pollutants in wastewater: Fabrication and characterization as well as photocatalytic activity evaluation” (Ms. catalysts-723305) has been revised again in response to the comments and suggestions.

    However, some testing data are still not done now, because a terrible coronavirus pneumonia has affected all scientific research institutions and universities in this period. I will provide these datas while I can go to work.  

    The other revised introduction is in enclosure.

Round 3

Reviewer 1 Report

The authors improved the manuscript within the possibilities (i.e. coronavirus spreading in China, facilities closed). Therefore, I suggest this paper for publishing

Author Response

Dear Reviewer

     We thanks very much for your comments to our paper (Ms. catalysts-723305). We have revised our paper in detail again.  The revised introduction are as following.  the revised parts were all noted as red words.

  1. I have added “tetrabutyl titanate”, “ of PP”, and deleted “excellent”, changed “real”, “superoxide free radical”, and “adds” for “nature”, “free radical”, and “improves”, respectively in the abstract part.
  2. In introduction part, I have changed some references, e.g. [5-6], [7], [8-9], [10-12], [13-15].
  3. The experimental part has been revised. I have changed “nanoparticles” and “performance ” for “powder” and “performances”, added “as-formed”, and “TiO2, TC”.Besides, I deleted “fractured”, corrected a mistake “photocatlyst” for “photocatalyst”.
  4. I have changed “decompose”, “real” for “degrade” and “nature” in page 4.
  5. In page 7, “behavior”, “rate”, and “is” have been changed for “behaviors”, “rates”, and “are” as well as references, e.g. [19], [19-20], and [7,11].
  6. I have added a financialsupport project “the key science and technology innovation demonstration projects of Henan province”, as seen in page 11.
  7. Some references have been changed, as seen in page 12.

Reviewer 2 Report

The revised manuscript could only be published after the authors restore the TEM measurement and the Cu leaching test (through ICP-OES or atomic absorption).

Moreover, regarding the detected XRD lines assigned to metallic Cu: the authors should check the literature and provide proper reference/s. It should be stated that there is only one XRD peak, not two.

Author Response

Dear Reviewer

     We thanks very much for your comments to our paper (Ms. catalysts-723305). We have revised our paper in detail again.  The revised introduction are as following.  the revised parts were all noted as red words.  

Point 1: The revised manuscript could only be published after the authors restore the TEM measurement and the Cu leaching test (through ICP-OES or atomic absorption).

Response 1: Thank you for giving us a chance, we will test the TEM images and atomic absorption spectra of all samples while our lab will open.

Point 2: Moreover, regarding the detected XRD lines assigned to metallic Cu: the authors should check the literature and provide proper reference/s. It should be stated that there is only one XRD peak, not two.

Response 2: I have check some references, I have changed “The two peaks at about 50° and 51° are possibly corresponded to metallic Cu of TC” has been changed for “The two peaks at about 43.5° and 50.6° possibly correspond to metallic Cu of TC [8, 18]”. I have prepared PA6/Cu composite, in that paper, the peaks of 43.5° and 50.6° corresponded metallic Cu, as seen reference 18. Besides, other researches have reported this results, e.g. “Mott D, Galkowski J, Wang L, et al. Synthesis of size-controlled and shaped copper nanoparticles[J]. Langmuir, 2007, 23(10): 5740-5745”.

In addition, the other revised introduction are as following, and the revised parts were all noted as red words.

  1. I have added “tetrabutyl titanate”, “ of PP”, and deleted “excellent”, changed “real”, “superoxide free radical”, and “adds” for “nature”, “free radical”, and “improves”, respectively in the abstract part.
  2. In introduction part, I have changed some references, e.g. [5-6], [7], [8-9], [10-12], [13-15].
  3. The experimental part has been revised. I have changed “nanoparticles” and “performance ” for “powder” and “performances”, added “as-formed”, and “TiO2, TC”.Besides, I deleted “fractured”, corrected a mistake “photocatlyst” for “photocatalyst”.
  4. I have changed “decompose”, “real” for “degrade” and “nature” in page 4.
  5. In page 7, “behavior”, “rate”, and “is” have been changed for “behaviors”, “rates”, and “are” as well as references, e.g. [19], [19-20], and [7,11].
  6. I have added a financialsupport project “the key science and technology innovation demonstration projects of Henan province”, as seen in page 11.
  7. Some references have been changed, as seen in page 12.
